# Prediction of Lymph Node Metastasis in T1 Colorectal Cancer Using Artificial Intelligence with Hematoxylin and Eosin-Stained Whole-Slide-Images of Endoscopic and Surgical Resection Specimens

**DOI:** 10.3390/cancers16101900

**Published:** 2024-05-16

**Authors:** Joo Hye Song, Eun Ran Kim, Yiyu Hong, Insuk Sohn, Soomin Ahn, Seok-Hyung Kim, Kee-Taek Jang

**Affiliations:** 1Department of Internal Medicine, Konkuk University Medical Center, Konkuk University School of Medicine, Seoul 05030, Republic of Korea; 20230514@kuh.ac.kr; 2Department of Medicine, Samsung Medical Center, Sungkyunkwan University School of Medicine, Seoul 06351, Republic of Korea; 3Department of R&D Center, Arontier Co., Ltd., Seoul 06735, Republic of Korea; insuks@gmail.com; 4Department of Pathology, Samsung Medical Center, Sungkyunkwan University School of Medicine, Seoul 06351, Republic of Korea; soomin17.ahn@samsung.com (S.A.); parmenides.kim@samsung.com (S.-H.K.); kt12.jang@samsung.com (K.-T.J.)

**Keywords:** lymph node metastasis, artificial intelligence, whole slide image, T1 colorectal cancer

## Abstract

**Simple Summary:**

We developed an attention-based whole slide image (WSI)-level classification deep learning model employing surgically and endoscopically resected specimens to predict LNM in T1 CRC. Our AI model with H&E-stained WSIs and without annotations showed good performance power with the validation of an independent cohort in a single center. The area under the curve of our model was 0.781–0.824, higher than that of previous artificial intelligence (AI) studies with only WSIs. Our AI model, which showed the highest sensitivity (92.9%), reduced unnecessary additional surgery by 14.2% more than using the current JSCCR guidelines (68.3% vs. 82.5%). This revealed the feasibility of using an AI model with only H&E-stained WSIs to predict LNM in T1 CRC.

**Abstract:**

According to the current guidelines, additional surgery is performed for endoscopically resected specimens of early colorectal cancer (CRC) with a high risk of lymph node metastasis (LNM). However, the rate of LNM is 2.1–25.0% in cases treated endoscopically followed by surgery, indicating a high rate of unnecessary surgeries. Therefore, this study aimed to develop an artificial intelligence (AI) model using H&E-stained whole slide images (WSIs) without handcrafted features employing surgically and endoscopically resected specimens to predict LNM in T1 CRC. To validate with an independent cohort, we developed a model with four versions comprising various combinations of training and test sets using H&E-stained WSIs from endoscopically (400 patients) and surgically resected specimens (881 patients): Version 1, Train and Test: surgical specimens; Version 2, Train and Test: endoscopic and surgically resected specimens; Version 3, Train: endoscopic and surgical specimens and Test: surgical specimens; Version 4, Train: endoscopic and surgical specimens and Test: endoscopic specimens. The area under the curve (AUC) of the receiver operating characteristic curve was used to determine the accuracy of the AI model for predicting LNM with a 5-fold cross-validation in the training set. Our AI model with H&E-stained WSIs and without annotations showed good performance power with the validation of an independent cohort in a single center. The AUC of our model was 0.758–0.830 in the training set and 0.781–0.824 in the test set, higher than that of previous AI studies with only WSI. Moreover, the AI model with Version 4, which showed the highest sensitivity (92.9%), reduced unnecessary additional surgery by 14.2% more than using the current guidelines (68.3% vs. 82.5%). This revealed the feasibility of using an AI model with only H&E-stained WSIs to predict LNM in T1 CRC.

## 1. Introduction

Colorectal cancer (CRC) is the second most fatal and the third most commonly diagnosed cancer worldwide [1,2]. However, CRC incidence and mortality have decreased due to colonoscopy screening, surveillance, and high-quality endoscopic treatment [3,4,5]. Endoscopic resection is recommended as the first-line treatment for early CRC without distant or lymph node metastasis (LNM). Although intramucosal CRC is not associated with LNM, submucosal CRC exhibits LNM in approximately 10% of cases [6,7,8,9,10]. Therefore, additional surgical resection is performed only when endoscopically resected specimens show high-risk features (deep submucosal (SM) invasion, lymphovascular invasion (LVI), tumor budding, or poorly differentiated histology) related to LNM [7,11,12,13,14,15]. However, even though additional surgery is recommended based on the current guidelines, LNM occurs in 2.1–25.0% of cases treated endoscopically and then surgically. In other words, 75–98% of additional surgeries are unnecessary [9,16,17,18,19]. The main challenge is to figure out LNM before undergoing surgery. Several studies have attempted to identify a method to predict LNM in patients with T1 CRC (tumor-invaded submucosa, according to the Japanese Society for Cancer of the Colon and Rectum (JSCCR) and American Joint Committee on Cancer) to reduce the number of unnecessary surgeries and minimize the risk of LNM. However, since low inter-observer agreement and limited indications of current guidelines, it is nearly impossible to predict LNM through pathologic examination based on Hematoxylin and eosin(H&E)-stained endoscopically resected specimen [20,21,22].

Recent studies have attempted to solve these problems using artificial intelligence (AI) [23,24,25,26,27]. The two strategical approaches for AI-assisted assessment of the risk of LNM in T1 CRC were pathologist-dependent and independent [28,29]. The pathologist-dependent strategy used text-based data, which included the histologic features obtained by a pathologist, such as depth of SM invasion, tumor differentiation, and LVI [23,25,30]. These test data-based AI models proved sufficient evidence with large cohorts and external validation, outperforming the current guidelines. Nevertheless, there were still limitations, such as varying pathologic criteria and standards among different guidelines and diagnostic disagreement among pathologists. To address these issues, a pathologist-independent AI model utilizing whole slide images (WSIs) has been reported. WSI-based AI models with hematoxylin and eosin staining alone, including our previous study, were simplified and less disruptive than current clinical best practices [26,27,31]. This strategy appears to be ideal for overcoming inter-observer discrepancy, but a relatively low area under the curve (AUC) compared to the pathologist-dependent method and external validation remain a challenge.

Even though our previous study demonstrated the potential of using an AI with H&E-stained WSIs from endoscopically resected specimens without handcrafted features to predict LNM in patients with T1 CRC, our model had certain limitations [31]. WSIs from endoscopically resected specimens had high-risk histological features of LNM because they belonged to patients who underwent additional surgery after endoscopic treatment. So, the previous model was unsuitable for predicting LNM in low-risk patients with T1 CRC. Also, the study population was small (*n* = 400), and AUC was relatively low. To increase the number of patients and WSIs from patients with low risk of LNM, we conducted a study with AI training and testing by expanding the scope to include previously endoscopically resected specimens from patients who underwent additional surgery due to the high risk of LNM, as well as surgical specimens from patients who underwent surgery for T1 CRC. Since the previous model lacked external validation with an independent cohort, we wanted to perform extensive external validation with WSIs from multi-centers and apply it to WSIs from T1 CRC patients who underwent endoscopic treatment. However, it takes a lot of time to prepare WSI from multicenter and obtain a 5-year overall survival rate in patients who have only undergone endoscopic treatment for T1 CRC. So, we conducted a study using an alternative method instead of external validation. To validate with an independent cohort in a single center, we aimed to develop an AI model with four versions comprising various combinations of training and test sets using H&E-stained WSIs from surgical and endoscopically resected specimens. Because endoscopic resected specimens contained only part of the SM layer while surgical specimens contained the entire layer of the intestine, the two cohorts were independent of each other. Additionally, we aimed to apply our AI program to predict LNM in T1 CRC samples.

## 2. Materials and Methods

### 2.1. Study Population 

The inclusion criteria were (1) patients who underwent surgical resection for newly diagnosed T1 CRC between 2003 and 2020 at the Samsung Medical Center or (2) patients who underwent endoscopic treatment including endoscopic mucosal resection (EMR) and endoscopic submucosal dissection (ESD) for newly diagnosed T1 CRC, and those who underwent additional surgery based on the JSCCR guidelines [14] due to high risk of LNM indicated by at least one of the following histologic features: positive resection margin, deep SM invasion (SM depth > 1 mm, Sm2/Sm3 for sessile T1, and Haggitt 4 for pedunculated T1 CRC), presence of LVI, poorly differentiated histology, or tumor budding: within 3 months after EMR/ESD from 2010 to 2018 at the Samsung Medical Center. The exclusion criteria were as follows: (1) unavailable H&E-stained slide, (2) unclear H&E-stained slide image for analysis, (3) no LN dissection, or (4) presence of synchronous invasive carcinoma. The study protocol was approved by the Institutional Review Board of Samsung Medical Center (2021-01-042-005). The requirement for informed consent from the patients was waived due to the use of de-identified data routinely collected during hospital visits.

### 2.2. Clinicopathologic Features and Preparation of Whole Slide Images for the Study Population 

Clinical data such as age at diagnosis, sex, body mass index, family history of CRC, presence of comorbidities, smoking status, alcohol consumption, and tumor location were reviewed. Additionally, pathologic features, such as the tumor size (length of cancer component measured by excluding adenoma component), positive resection margin, depth of SM invasion, LVI, histologic differentiation (based on the least differentiated component), tumor budding, and microsatellite instability, were reviewed by a pathologist and used only for comparison with the predictive performance of our model. Assessment of histological differentiation was based on the least high-grade pattern of the carcinoma, which often co-exists with dominant elements of low-grade patterns. Immunostaining with D2–40 was occasionally performed for lymphatic vessel to determine whether it was a true lymphatic or an iatrogenic empty space caused by tissue being pushed in the process of specimen fixing in formaldehyde and making into slide. 

Surgical and endoscopically resected specimens were fixed in formaldehyde and embedded in paraffin. Tissue specimens were cut into sections with 3 μm that were placed on the slides. During preparation, the artifact was removed from ethanol and a 50 °C floating hot water tank. H&E-stained specimen slides were scanned using a VENTANA iScan HT scanner (Roche Diagnostics, Basel, Switzerland) at ×20 magnification.

### 2.3. Deep Learning Artificial Intelligence Model Development

The deep learning method used in this study is the same as that employed in our previous study [31]. We developed an attention-based WSI-level classification deep learning model to predict whether a WSI is LNM positive or negative (Figure 1). The model was trained for a binary classification task, where the input was a WSI, and the output was the probability of the WSI being LNM-positive.

The model is an end-to-end neural network comprising a deep convolutional neural network (DCNN), attention module (AM), and classification module (CM) [32,33]. The DCNN was pre-trained with patches labeled as positive (patches from LNM positive WSIs) or negative (patches from LNM negative WSIs) to learn features in histopathological images in advance and to function as a patch image feature extractor (FE) [34]. The AM computes an attention score (AS), between 0 and 1, for each patch image in a WSI; the sum of these scores is equal to 1. An attention mechanism was used to visualize the spatial distribution of ASs of the WSIs. A higher AS indicates that the patch image is relatively more informative and has a greater influence on the final classification decision.

The model’s inferencing details are as follows: For a given WSI, all tissue regions are patched in a tiling manner and used as input for the DCNN FE, which compresses and encodes each patch image into a 512-dimensional feature vector (FV). The FVs are further aggregated into a single WSI-level FV (WSI deep feature) with 512 dimensions using their weighted average determined by the AS, computed by the AM [32]. The final WSI deep feature is then input into the CM to obtain the final prediction for LNM.

### 2.4. Statistical Analysis

Continuous variables are expressed as medians with interquartile ranges (IQRs) and analyzed using Student’s *t*-test and the Mann–Whitney U test. Statistical significance was set at *p* < 0.05. All statistical analyses were performed using SPSS software version 28 for Windows (SPSS Inc., Chicago, IL, USA). 

The AI performance was evaluated using the AUC receiver operating characteristic curve (ROC). ROC is a probability curve, and AUC represents the degree or measure of separability. It showed how much the model was capable of distinguishing between classes.

To validate our model in case of a lack of other hospital WSI, we developed a model with four versions of training and test set combinations using endoscopically and surgically resected specimens: Version 1, Train and Test: surgical specimens; Version 2, Train and Test: endoscopic and surgically resected specimens; Version 3, Train: endoscopic and surgical specimens and Test: surgical specimens; and Version 4, Train: endoscopic and surgical specimens and Test: endoscopic specimens. 

We performed a five-fold cross-validation (CV) on the training set, preserving the percentage of each class to determine how well our approach worked on each fold. Consequently, each of the five models trained in CV was applied to the held-out test set, and the results were obtained by averaging the output predictions. By comparing our AI model with a model using clinicopathological features, we trained a random forest (RF) classifier with 500 trees to predict LNM [35]. RF is a versatile and widely used machine learning algorithm that constructed multiple decision trees and combined their outputs for robust and accurate predictions.

The optimal cut-off sensitivity and specificity of each model were evaluated using the Youden index, the maximum potential effectiveness of a diagnostic biomarker, and a common summary measure of the ROC curve [36]. And we used McNemar’s tests, non-parametric test used to analyze paired nominal data, to compare predictive performances between our model and JSCCR guidelines, the most widely used guidelines in Asia.

## 3. Results

### 3.1. Baseline Characteristics of the Study Population

A total of 1737 patients with T1 CRCs (1046 surgical resections and 691 endoscopic resections followed by surgery) were eligible for this study, and 456 patients were excluded. Thus, 1281 patients (881 surgical resections and 400 endoscopic resections followed by surgery) were analyzed (Figure 2). Their baseline clinicopathological characteristics are presented in Table 1. The median age at CRC diagnosis was much younger in patients with endoscopic resection followed by additional surgery (59.0; IQR, 52.0–65.0) than in patients with surgical resection (60.0; IQR, 52.0–69.0). Men accounted for 59.6% of the total population. The percentage of patients with a family history of CRC, ex-/current smoker, or alcohol ex-/current drinker was higher in endoscopic resection followed by additional surgery. Patients without a family history of CRC accounted for 89.2% of the patients. High-risk pathologic features related to LNM, including LVI, tumor budding, positive resection margin, and microsatellite instability, were more in patients with endoscopic resection followed by additional surgery than surgical resection. 

Patients with CRC and LNM accounted for 6.6% (*n* = 58) of patients with surgical resection and 17.8% (*n* = 71) of patients with endoscopic resection followed by additional surgery. LN yield, the total number of LNs retrieved after surgery was 22,022. The LN ratio, the ratio of positive LNs out of the total removed, was 1.24% (273/22,022). In our study, an average of 17 LNs were retrieved in each surgery. When we compared the past group (patients who underwent surgery in 2003–2010) and the recent group (2011–2020), an average of 16 LNs were retrieved per surgery in the past group, and 18 LNs were retrieved per surgery in the recent group.

### 3.2. Train and Test Set in Model with Four Versions 

A total of 2604 WSIs (184 positive LNM and 1139 negative LNM) from 881 surgical specimens (102 positive LNM and 791 negative LNM) and 400 endoscopically resected specimens (82 positive LNM and 348 negative LNM) were used to develop the model. A summary of the four versions is presented in Table 2. 

In Version 1, 893 WSIs (102 positive LNM and negative 791 LNM) from 881 surgical specimens were randomly split into training and test sets in a ratio of 4:1 at the patient level. Accordingly, 80 and 21 patients with positive LNM and 624 and 156 with negative LNM were assigned to the training and test sets, respectively. 

In Version 2, 1323 WSIs (184 positive LNM and negative 1139 LNM) from 881 surgical specimens and 400 endoscopically resected specimens were randomly split into training and test sets in a ratio of 4:1 at the patient level. Accordingly, 137 and 35 patients with positive LNM and 887 and 222 patients with negative LNM were assigned to the training and test sets, respectively. 

In version 3, 1144 WSIs (165 positive LNM and negative 1068 LNM) from 881 surgical specimens and 400 endoscopically resected specimens were randomly split into the training set and surgical specimens were randomly split into the test set in a ratio of 6:1 at the patient level. Accordingly, 137 and 21 patients with positive LNM and 887 and 156 patients with negative LNM were assigned to the training and test sets, respectively. 

In Version 4, 1145 WSIs (163 positive LNM and 982 negative LNM) from 881 surgical specimens and 400 endoscopically resected specimens were randomly split into the training set and endoscopically resected specimens were randomly split into the test set in a ratio of 13:1 at the patient level. Accordingly, 137 and 14 patients with positive LNM and 887 and 66 patients with negative LNM were assigned to the training and test sets, respectively.

### 3.3. Area under the Curve for Predicting Lymph Node Metastasis

The AUCs for predicting LNM in T1 CRC using the AI model with histopathological images of endoscopic and surgical specimens and RF with clinicopathological features are shown in Table 3. Our model showed better prediction performance, with an AUC of 0.758–0.830 for the training set and 0.781–0.824 for the test set, than that of the model with clinicopathological features (AUC 0.516–0.683 in the test set). The ROC curves of the AI model are shown in Figure 3. The sensitivity and specificity of each version were 71.4% and 92.9% for version 1, 71.4% and 84.2% for version 2, 76.2% and 85.9% for version 3, and 92.9% and 57.6% for version 4 respectively. 

### 3.4. Predictive Performance of Model with Four Versions vs. That of JSCCR Guidelines

We compared the performance of our model (four versions) with that of JSCCR guidelines using the test set (Table 4). JSCCR guidelines recommend that additional colorectal surgery be performed when endoscopically resected specimens show at least one of the high-risk features of LNM. JSCCR guideline did not allow any LNM of T1 CRC, which resulted in unnecessary additional surgery. It meant that this strategy showed 100% sensitivity and 0% specificity. On the other hand, to reflect reality as much as possible, we used the sensitivity and specificity of our model determined through the Youden index instead of setting it at 100% sensitivity to not allowing false negatives. The rate of unnecessary additional surgery attributable to misdiagnosing patients with negative LNM as having positive LNM was anticipated to be from 42.3% to 68.3% by our model with four versions and 82.5 to 88.1% by the JSCCR guidelines. Based on the results of the analysis, our model avoided at least 14.2% of unnecessary additional surgeries than predicted using the current JSCCR guidelines. 

### 3.5. Attention Score

The attention mechanism interprets the effect of each patch on the final WSI-level decision using a scoring system. The calculated ASs of WSIs for positive LNM are displayed as a heatmap, highlighting regions of interest (ROIs), where ASs were normalized using a simple min-max normalization method (Figure 4). Sample patch images of LNM-positive WSIs are shown in Figure 4B. The prominent features of the sample patch images with high ASs were tumor budding and micropapillary patterns.

## 4. Discussion

The risk of LNM in T1 CRC is associated with the following histological risk factors in endoscopically resected specimens: LVI, tumor budding, histological grade, and depth of SM invasion. In cases of high-risk LNM, additional surgery is recommended based on the current guidelines. However, the risk of LNM in T1 CRC after additional surgery with LN dissection is estimated to be 6–14% [37]. Therefore, to avoid unnecessary surgery, it is important to predict LNM using endoscopically resected specimens before surgery. However, it is almost impossible to predict LNM based on pathological evaluation using only H&E-stained, endoscopically resected specimens. To address this issue, several recent studies have used AI models to determine the histological risk of LNM [23,24,26,38,39]. However, these DL models are still in the early stages of development and require extensive external validation [28,29,40].

In our previous study, we developed a prediction model analyzing H&E-stained WSIs for LNM in T1 CRC using DL without manual pixel-level annotation. Compared to existing studies, this study had the strengths of a relatively large target group that underwent additional surgery after endoscopic resection, resulting in better AUCs for predicting LNM with H&E-stained WSI information alone. Although the previous study included high-risk histological features of LNM, the absolute number of enrolled patients was small. Moreover, as it was difficult to perform an external validation using scanned WSIs from other hospitals in a relatively short time, no external validation was performed. Consequently, we planned to validate our model using not only endoscopic specimens but also surgical specimens. To increase the number of patients and WSIs from patients with low risk of LNM, we included patients who underwent surgery between 2003–2020, a longer study period than those who underwent endoscopic resection followed by surgery (2010–2018). Massive surgical specimens that were performed when surgery was the only treatment option for T1 CRC showed a lower risk of LNM compared to endoscopic resected specimens. Indeed, pathologic characteristics of patients with surgical resection showed a lower risk of LNM than in patients with endoscopic resection, followed by surgery. Therefore, as an alternative to the independent cohort, we trained and tested the model in four versions and assessed its performance.

We developed an AI model with a DL program to predict LNM in T1 CRC using surgical and endoscopic specimens. The AUCs of our DL model were 0.758–0.830 for the training set and 0.781–0.824 for the test set, which were improved compared with the previous model (AUC: 0.747 in training and 0.767 in testing sets). This could be explained by the fact that surgical specimens contained much more tissue and information than endoscopically resected specimens, and over 1000 WSIs were used for analysis compared with previous studies. Among four versions, Version 3 (train: endoscopic and surgical specimens, test: surgical specimens), which contained surgical specimens in both training and test sets, had the highest performance power. However, all versions of the AI model showed acceptable AUC ranges for predicting LNM in patients with T1 CRC. On the other hand, RF with clinicopathological features showed a lower AUC (0.516–0.701) than the AI model. Only Version 1 (training and test: surgical specimen) had an AUC barely greater than 0.7. 

Among four versions, Version 4 (train: endoscopic and surgical specimens, test: endoscopic specimens) was the closest to the actual prediction target. Because the ultimate goal of the AI model was to predict LNM in patients who were only treated with endoscopic treatment for T1 CRC, however, considering the study results, it could be assumed that test with endoscopic resected specimen was difficult to predict LNM. The AUC of RF with clinicopathologic features was the lowest (0.516) in Version 4. The AUC of Version 4 was the lowest among the four versions, even though the difference was not significant. However, it was remarkable that the AI model with Version 4 had the greatest improvement of AUC (0.265), compared to RF with clinicopathologic features, among the four versions. Thus, our newly developed model showed the possibility of application in clinical practice for LNM prediction. 

In a previous study, when we compared the AI model with JSCCR guidelines, we ideally set the cutoff threshold of the AI model at 100% sensitivity not to allow missed LNM, like JSCCR guidelines. As a result, the previous model reduced unnecessary additional surgeries by 15.1% than the current JSCCR guidelines. However, setting the cutoff threshold at 100% sensitivity might not reflect reality, so we used the Youden index for setting the cutoff in our present study. The present model avoided 14.2–45.8% of unnecessary additional surgeries than predicted using the current JSCCR guidelines while allowing missed LNM, which ranged from 7.1 to 28.6%. The AI model that reduced unnecessary additional surgery allowed more missed LNM. So, careful interpretation of results was needed. Considering, acceptable the lowest rate of missed LNM, the ultimate target population, and the improvement of AUC using the AI model, the AI model with Version 4 was compatible with clinical practice to predict LNM in T1 CRC. It reduced 14.2% of unnecessary additional surgeries than predicted using the current JSCCR guidelines while allowing 7% of missed LNM. 

Previous studies using AI to determine the risk of LNM on histology showed AUC ranging between 0.567 and 0.938, consistent with our results (0.781–0.824) [26,27,31,38,39,41]. Most studies included clinicopathologic variables and/or additional immunohistochemistry performed by pathologists. However, several recent studies have demonstrated the potential of applying DL to predict LNM in T1 CRC using H&E-stained WSIs without a histological assessment [26,27,31]. This pathologist-independent strategy may be the focus of the next era of T1 CRC management [29]. Similarly, our AI model also used WSIs without annotation and included a large number of T1 CRC cases (*n* = 1281). The model was validated using an independent cohort at the same institution and showed the best AUC among the AI models using only WSI. 

In contrast to previous studies, Kasahara et al. developed an AI model with 80–85% accuracy using biopsy specimens and mucosal layer of the surgical specimens in the absence of biopsy specimens to predict patients with T1 CRC without LNM and their LNM risk before treatment, and select appropriate procedure before treatment [41]. The study suggests that biopsy specimen characteristics are associated with LNM risk. However, they implemented a weakly supervised model with a small number of patients and images of the site of choice for each ROI selected by the pathologist. Our previous study was conducted using high-risk histological features of LNM and was unsuitable for predicting LNM in low-risk patients with T1 CRC. However, 10.1% of the present study population had LNM, consistent with previous Japanese studies on long-term outcomes of CRC (10.8–12.4%) [42,43]. Additionally, the sensitivity and specificity of the present model were improved compared to our previous study. Therefore, it might be acceptable to apply our AI model to predict LNM in patients with low-risk T1 CRC. 

Even though depth of SM invasion and tumor budding were risk factors of LNM, the meaning of these is controversial, and inter-observer variation for measurement existed. According to JSCCR guidelines, SM depth >1000 μm was one of the risk factors for LNM, but several studies showed differences in SM invasion depth related to LNM [44,45,46]. The International Tumor Budding Consensus Conference guidelines recommended the use of a three-tier system for risk stratification: Bd 1, low budding (0–4 buds); Bd 2, intermediate budding (5–9 buds); Bd 3, high budding (10 or more buds) [47]. In p1 CRC, Bd 2 and Bd 3 were associated with an increased risk of LNM, whereas in stage II CRC, Bd 3 is associated with an increased risk of recurrence and mortality. Moreover, the current tumor bud assessment system focused only on the tumor bud count and did not account for other features [48]. So, we wanted to figure out the meaning of depth of invasion for LNM and tumor budding using an attention-based WSI-level classification deep learning model. Patch images with high ASs appeared to be located in the transformation zone, which is the boundary between normal and cancerous tissues. In our study, the prominent features of the patch images with high ASs were poorly differentiated histological grades, tumor budding and micropapillary patterns, well-known pathological factors associated with poor prognosis [49,50]. Additionally, Brockmoeller et al. demonstrated an association between inflamed fat in CRC and LNM and that AI had the potential to discover new mechanisms in cancer progression [26]. However, unlike their study, our patch images with high AS did not contain inflamed fat. 

Nevertheless, our study had some limitations. First, it was a single tertiary center retrospective study, which has a potential for bias. Second, the AUC of our method was relatively low to use as a prediction model despite the improvement achieved by increasing the study population and using various training and test sets for endoscopic and surgical specimens. Additionally, the DL model was influenced by class balance, and intrinsic LNM in T1 CRC was low. Nonetheless, among the AI models that used only WSIs, our study showed the best performance. Third, extensive external validation was lacking. We performed validation with an independent cohort in a single center; however, staining using H&E, variations in WSI color, and clinicopathological characteristics in a single center were similar. Therefore, the discriminating power may have been overestimated [29,40]. However, our study enrolled a relatively large number of patients compared with previous studies. Moreover, the clinicopathological features did not seem to be important in predicting LN metastasis in our AI model with H&E-stained WSI. Finally, our model was not validated in a cohort that underwent endoscopic resection of T1 CRC without additional surgery, which is a part of the model’s real target. 

## 5. Conclusions

In conclusion, our AI model with H&E-stained WSIs and without pathologists showed higher performance power (AUC, 0.782–0.824) with validation of an independent cohort in a single center than previous studies. Since WSIs from 1281 patients with low to high risk of LNM were used to develop the present AI model, it was suitable for predicting LNM even in low-risk patients with T1 CRC. Moreover, this model reduced 14.2% of unnecessary additional surgeries than predicted using the current JSCCR guidelines while allowing 7% of missed LNM. This revealed the feasibility of using an AI model with only H&E-stained WSIs to predict LNM in T1 CRC. However, to apply our model to real-world clinical practice, extensive external validation with WSI from multiple centers and patients who undergo only endoscopic treatment is warranted.

## Figures and Tables

**Figure 1 cancers-16-01900-f001:**
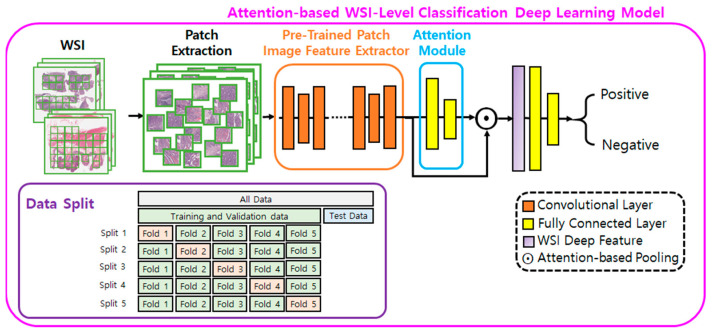
Pipeline of the approach for classifying lymph node metastasis. WSI, whole slide image.

**Figure 2 cancers-16-01900-f002:**
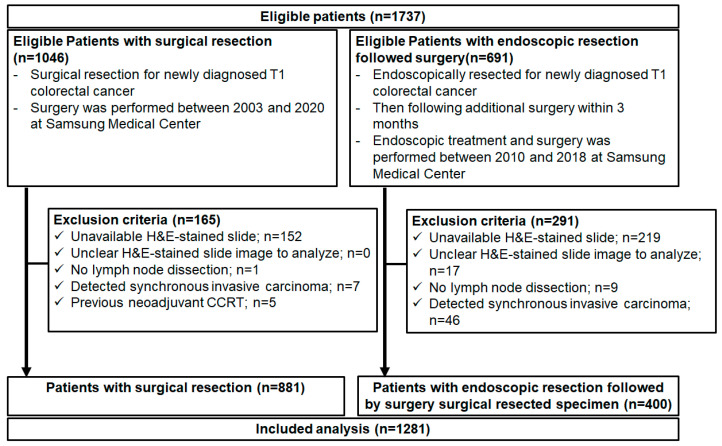
Flow chart of study population. CCRT, concurrent chemoradiotherapy.

**Figure 3 cancers-16-01900-f003:**
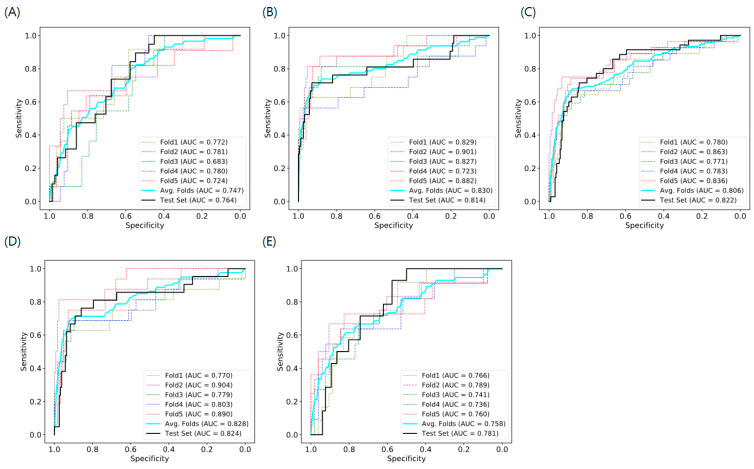
Area under the ROC curve for attention-based WSI-level classification deep learning model for predicting lymph node metastasis in T1 colorectal cancer. (**A**) Previous model, (**B**) Version 1, (**C**) Version 2, (**D**) Version 3, (**E**) Version 4 ROC, receiver operating characteristic; WSI, whole-slide image; Version 1, train and test: surgical specimen; Version 2, train and test: endoscopic and surgical specimen; Version 3, train: endoscopic and surgical specimens and test: surgical specimen; Version 4, train: endoscopic and surgical specimens and test: endoscopic specimen; AUC: area under the curve; Avg: average.

**Figure 4 cancers-16-01900-f004:**
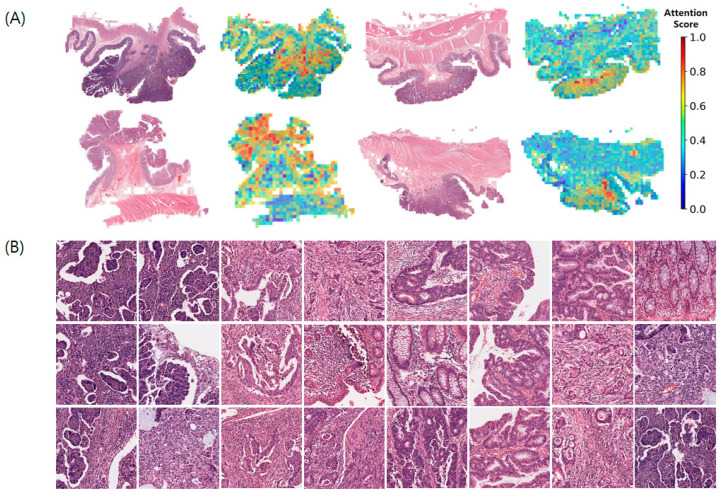
Attention score visualization in WSIs of positive lymph node metastasis. (**A**) Spatial distribution of attention scores on the WSIs, (**B**) Patch-images stratified by high attention scores WSIs, whole slide images.

**Table 1 cancers-16-01900-t001:** Baseline characteristics of study population.

			Endoscopic Resection Followed by Additional Surgery (*n* = 400)		Surgical Resection (*n* = 881)		
		Total	Negative LNM (*n* = 329)	Positive LNM (*n* = 71)	Negative LNM (*n* = 780)	Positive LNM (*n* = 101)	*p* Value *
Clinical features							
Age at diagnosis	Year (IQR)	60.0 (52.0–68.0)	59.0 (52.0–65.0)	60.0 (52.0–68.0)	60.0 (52.0–69.0)	59.0 (52.0–67.0)	0.041
Sex	Male	764 (59.6)	193 (58.7)	46 (64.8)	467 (59.9)	58 (57.4)	0.504
	Female	517 (40.4)	136 (41.3)	25 (35.2)	313 (40.1)	43 (42.6)	
Body mass index	kg/m^2^ (IQR)	24.1 (22.2–26.1)	23.9 (22.0–26.1)	24.8 (23.3–27.4)	24.1 (22.2–26.0)	24.8 (23.0–26.0)	0.704
Presence of comorbidity	No	795 (62.1)	217 (66.0)	36 (50.7)	477 (61.2)	65 (64.4)	0.299
	Yes	486 (37.9)	112 (34.0)	35 (49.3)	302 (38.7)	36 (35.6)	
Family history of CRC	No	1143 (89.2)	294 (86.3)	61 (85.9)	708 (90.8)	90 (89.1)	0.014
	yes	138 (10.8)	45 (13.7)	10 (14.1)	72 (9.2)	11 (10.9)	
Smoking status	No	912 (71.2)	214 (65.0)	43 (60.6)	588 (75.4)	67 (66.3)	<0.001
	Ex-smoker	201 (15.7)	59 (17.9)	10 (14.1)	104 (13.3)	23 (22.8)	
	Yes	168 (13.1)	56 (17.0)	18 (25.4)	88 (11.3)	11 (10.9)	
Alcohol consumption	No	809 (63.2)	192 (58.4)	31 (43.7)	523 (67.1)	63 (62.4)	<0.001
	Ex-drinker	71 (5.5)	27 (8.2)	9 (12.7)	29 (3.7)	6 (5.9)	
	Yes	401 (31.3)	110 (33.4)	31 (43.7)	228 (29.2)	32 (21.7)	
Tumor location	Left side	913 (71.3)	241 (73.3)	50 (70.4)	542 (69.5)	80 (79.2)	0.236
	Right side	368 (28.7)	88 (26.7)	21 (29.6)	238 (30.5)	21 (20.8)	
Pathologic features							
Size of cancer	mm (IQR)	15.0 (10.0–22.0)	10.0 (7.3–14.0)	8.0 (7.0–12.0)	20.0 (15.0–25.0)	16.5 (14.3–25.0)	<0.001
Depth of SM invasion	μm (IQR)	1775.0	1800.0	1500.0			N/A
		(1000.0–2200.0)	(1075.0–2300.0)	(1000.0–2000.0)			
	SM1	340 (38.6)			318 (40.8)	22 (21.8)	N/A
	SM2	218 (24.7)			185 (23.7)	33 (32.7)	
	SM3	323 (36.7)			277 (35.5)	46 (45.5)	
Differentiation	Well	760 (59.3)	182 (55.3)	55 (77.5)	480 (61.5)	43 (42.6)	0.210
	Moderate	485 (37.9)	133 (40.4)	14 (19.7)	286 (36.7)	52 (51.5)	
	Poorly	36 (2.8)	14 (4.3)	2 (2.8)	14 (1.8)	6 (5.9)	
Lympho-vascular invasion	No	1030 (80.4)	243 (73.9)	49 (69.0)	691 (89.6)	47 (46.5)	<0.001
	Yes	251 (19.6)	86 (26.1)	22 (31.0)	89 (11.4)	54 (53.5)	
Tumor budding	No	1084 (84.6)	289 (87.8)	62 (87.3)	667 (85.5)	66 (65.3)	0.021
	Yes	197 (15.4)	40 (12.2)	9 (12.7)	113 (14.5)	35 (34.7)	
Positive resection margin	No	1167 (91.1)	235 (71.4)	51 (71.8)	780 (100)	101 (100)	<0.001
	Yes	114 (8.9)	94 (28.6)	20 (28.2)	0	0	
Microsatellite stability	Stable	915 (71.5)	82 (25.0)	28 (39.4)	667 (85.5)	89 (88.1)	<0.001
	Unstable	86 (6.7)	8 (2.4)	2 (2.8)	69 (8.8)	7 (6.9)	
	Unknown	279 (21.8)	238 (72.6)	41 (57.7)	44 (5.6)	5 (5.0)	

* *p*-value: difference between endoscopic resection followed by additional surgery (*n* = 400) and surgical resection (*n* = 881). LN, lymph node; IQR, interquartile range; CRC, colorectal cancer; SM, submucosal; N/A; not applicable; SM1, upper third, if depth of submucosal invasion was pragmatically divided in equal thirds according to Kudo classification; SM2, middle third; SM3, lower third.

**Table 2 cancers-16-01900-t002:** Composition of number of patients and WSI according to lymph node metastasis in the training and test set of artificial intelligence model with four versions.

	LNM		Previous Study	Version 1	Version 2	Version 3	Version 4
Training (5 fold) set	+	No of patients	57	80	137	137	137
	+	No of WSI	63	81	144	144	144
	−	No of patients	263	624	887	887	887
	−	No of WSI	277	634	911	911	911
Test set	+	No of patients	14	21	35	21	14
	+	No of WSI	19	21	40	21	19
	−	No of patients	66	156	222	156	66
	−	No of WSI	71	157	228	157	71

WSI, whole slide image; LNM, lymph node metastasis; Version 1, train and test: surgical specimen; Version 2, train and test: endoscopic and surgical specimen; Version 3, train: endoscopic and surgical specimen and test: surgical specimen; Version 4, train: endoscopic and surgical specimen and test: endoscopic specimen.

**Table 3 cancers-16-01900-t003:** Area under the curve of artificial intelligence model, compared to random forest with clinicopathologic features predicting of lymph node metastasis in in T1 CRC.

	Cross-Validation on Train Set	Previous Study	Version 1	Version 2	Version 3	Version 4
Attention-base WSI-level classification deep learning model	1	0.772	0.829	0.780	0.770	0.766
	2	0.781	0.901	0.863	0.904	0.789
	3	0.683	0.827	0.771	0.779	0.741
	4	0.780	0.723	0.783	0.803	0.736
	5	0.724	0.882	0.836	0.890	0.760
	Average of five-folds	0.747	0.830	0.806	0.828	0.758
	Test set	0.764	0.814	0.822	0.824	0.781
RF with clinicopathologic features *	1	0.598	0.659	0.653	0.728	0.512
	2	0.574	0.713	0.722	0.704	0.712
	3	0.703	0.710	0.739	0.728	0.746
	4	0.631	0.729	0.725	0.712	0.721
	5	0.623	0.670	0.647	0.666	0.593
	Average of five-folds	0.626	0.696	0.697	0.708	0.657
	Test set	0.598	0.701	0.635	0.683	0.516

WSI, whole slide image; RF, random forest; Version 1, train and test: surgical specimen; Version 2, train and test: endoscopic and surgical specimen; Version 3, train: endoscopic and surgical specimens and test: surgical specimen; Version 4, train: endoscopic and surgical specimens and test: endoscopic specimen. * Clinicopathological features included age at diagnosis, sex, body mass index, presence of comorbidities, family history of CRC, smoking status, alcohol consumption, tumor location, size of cancer, depth of submucosal invasion, lymphovascular invasion, histologic differentiation, tumor budding, and microsatellite instability.

**Table 4 cancers-16-01900-t004:** Predictive value of our artificial intelligence model with four versions and JSCCR guideline for lymph node metastasis in patients with T1 colorectal cancer.

		Artificial Intelligence	JSCCR	*p* Value
Version 1	Sensitivity (%)	71.4	100	<0.001
	Specificity (%)	92.9	0	<0.001
	PPV (%)	57.7	11.9	<0.001
	Accuracy (%)	90.4	11.9	<0.001
	Unnecessary additional Surgery (%)	42.3	88.1	<0.001
	Missed LNM (%)	28.6	0	<0.001
	Reduced unnecessary additional surgery (%) *	45.8		
Version 2	Sensitivity (%)	71.4	100	<0.001
	Specificity (%)	84.2	0	<0.001
	PPV (%)	41.7	13.6	<0.001
	Accuracy (%)	82.5	13.6	<0.001
	Unnecessary additional Surgery (%)	58.3	86.4	<0.001
	Missed LNM (%)	28.6	0	<0.001
	Reduced unnecessary additional surgery (%)	28.1		
Version 3	Sensitivity (%)	76.2	100	<0.001
	Specificity (%)	85.9	0	<0.001
	PPV (%)	42.1	11.9	<0.001
	Accuracy (%)	84.7	11.9	<0.001
	Unnecessary additional Surgery (%)	57.9	88.1	<0.001
	Missed LNM (%)	23.8	0	<0.001
	Reduced unnecessary additional surgery (%)	30.2		
Version 4	Sensitivity (%)	92.9	100	<0.001
	Specificity (%)	57.6	0	<0.001
	PPV (%)	31.7	17.5	<0.001
	Accuracy (%)	63.8	17.5	<0.001
	Unnecessary additional Surgery (%)	68.3	82.5	<0.001
	Missed LNM (%)	7.1	0	<0.001
	Reduced unnecessary additional surgery (%)	14.2		

* Reduced unnecessary additional surgery when using the artificial intelligence model, compared to JSCCR guidelines. JSCCR, Japanese Society for Cancer of the Colon and Rectum; Version 1, train and test: surgical specimen; Version 2, train and test: endoscopic and surgical specimen; Version 3, train: endoscopic and surgical specimens and test: surgical specimen; Version 4, train: endoscopic and surgical specimens and test: endoscopic specimen; PPV, positive predictive value; LNM, lymph node metastasis

## Data Availability

The raw data supporting the conclusions of this article will be made available by the authors on request.

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
