# Peer review of "Prediction of Lymph Node Metastasis in T1 Colorectal Cancer Using Artificial Intelligence with Hematoxylin and Eosin-Stained Whole-Slide-Images of Endoscopic and Surgical Resection Specimens"

_cancers, 2024, doi:10.3390/cancers16101900_

Round 1

Reviewer 1 Report

Comments and Suggestions for Authors

1. The article is highly similar to the following article previously published by the authors. This can be seen with the Ithenticate report. The similarity status should be reviewed again.

"https://doi.org/10.1007/s00535-022-01894-4"

2. The originality aspect of the article is weak and a very similar approach to previously published articles is presented.

3. Although many articles are cited in the introduction, it is weak in terms of content in addressing the problem.

4. The conclusion is weak. More detailed informative and guiding information should have been given in this section.

Author Response

Point-by-point response

We sincerely thank you for the opportunity to revise our manuscript entitled “Prediction of lymph node metastasis in T1 colorectal cancer using artificial intelligence with hematoxylin and eosin-stained whole slide images of endoscopic and surgical resection specimens" (cancers-2980766) to allow it to be reconsidered for publication in Cancers. We have revised the manuscript based on the reviewers’ helpful comments and recommendations. We have done our best to address to these comments thoroughly, and our responses are as follows:

Answers to comments from the Editorial team:

Reviewer #1:

  1. The article is highly similar to the following article previously published by the authors. This can be seen with the Ithenticate report. The similarity status should be reviewed again. https://doi.org/10.1007/s00535-022-01894-4

à Author Response: Thank you for your thoughtful comment. The final goal of our AI study is to apply the model to real-world clinical practice to predict lymph node metastasis (LNM) by minimizing the risk of LNM. To achieving this goal, extensive external validation with whole slide images (WSIs) from multi-centers is required and AI model should be applied to WSI from T1 CRC patients who underwent only endoscopic treatment. However, considering that it took a lot of time to prepare WSI from multicenter and obtain 5-year overall survival rate in patients who underwent only endoscopic treatment for T1 CRC, we thought that conducting the study with alternative method instead of external validation might be helpful and essential for achieving ultimate goal. We totally agreed with your comment that present study was conducted using the AI model developed based on previous experience, so it might look similar to previous study. However, we refined the present AI model using previous model with different way for training and test and using independent cohort. While endoscopic resected specimens contained only mucosa and submucosa layer, surgical specimens contained full layer of intestinal wall and lymph node. Indeed, our present study was different from previous study. Following your advice, we had revised our manuscript through emphasizing the difference between present and previous study and reinforcing the strength of our present study, distinguished from other studies.

  1. The originality aspect of the article is weak and a very similar approach to previously published articles is presented.

à Author Response: Thank you for your kind comment. As I mentioned at response #1, the final goal of our AI study is to apply the model to real-world clinical practice to predict lymph node metastasis (LNM) by minimizing the risk of LNM. To achieving this goal, extensive external validation with whole slide images (WSIs) from multi-centers is required and AI model should be applied to WSI from T1 CRC patients who undergo only endoscopic treatment. However, considering that it took a lot of time to prepare WSI from multicenter and obtain 5-year overall survival rate in patients who underwent only endoscopic treatment for T1 CRC, we thought that conducting the study with alternative method instead of external validation might be helpful and essential for achieving ultimate goal.

Even though, it would be ideal to develop a new model every time for each study, unfortunately developing a model required a lot of time and clinical data. So, we wanted to refine previous model using different training and test methods and independent cohort. Similar to our studies, Kudo et al. performed their next study, which included endoscopic resection alone cases with additional external validation to overcome the limitation of previous their study. (Gastroenterology, 2021, 160 (4); 1075-0184) Since, our study aimed to develop pathologist-independent AI model using only WSIs, we though that AI model based on similar approach but trained and tested with different WSI (endoscopically resected specimen vs. surgical specimen), was different from previous model. Because the performance of AI model varied depending on which test sets were used, fitting and validation of AI model were always important for developing AI model. We reinforced our model with using WSIs from over 1000 patients and not only endoscopic resected specimen but also surgical specimens. As mentioned at discussion section, endoscopic resected specimens had high risk histological features of LNM, because it belonged to patients who underwent additional surgery after endoscopic treatment. So, previous model was unsuitable for predicting LNM in low-risk patients with T1 CRC. However, since present study included massive surgical specimens that performed when surgery was the only treatment option for T1 CRC, it could be applied to low to high risk T1 CRC. Indeed, patients of previous study showed a relatively higher rate of LNM (17.8%) than previous Japanese studies with long-term outcomes (10.8–12.4%). On the other hand, patients in present study showed 13.4 % of LNM. Moreover, to validate with independent cohort in single center, we developed an AI model with four versions comprising various combinations of training and sets using WSIs from surgical and endoscopically resected specimens. As a result, present model showed better performance with AUC 0.824 and reduced at least 14.2% of unnecessary additional surgery in clinical setting, not ideal setting (set the cutoff threshold at 100% sensitivity). Therefore, it revealed that possibility of application in clinical practice for LNM prediction. Of course, we have prepared extensive external validation with collecting WSI from multi-center and following up long-term periods with multicenter cohort those who underwent only endoscopic treatment for T1 CRC. Following your advice, we had revised our manuscript through emphasizing the difference between present and previous study and reinforcing the strength of our present study, distinguished from other studies.

  1. Although many articles are cited in the introduction, it is weak in terms of content in addressing the problem.

à Author Response: Thank you for your considerable comment. Following your advice, we reinforced introduction section.  

Introduction section (3-4 page)

Colorectal cancer (CRC) is the second most fatal and the third most commonly diagnosed cancer worldwide. [1, 2] However, CRC incidence and mortality have decreased due to colonoscopy screening, surveillance, and high-quality endoscopic treatment Endoscopic resection is recommended as the first-line treatment for early CRC without distant or lymph node metastasis (LNM). Although intramucosal CRC is not associated with LNM, submucosal CRC exhibits LNM in approximately 10% of cases. Therefore, additional surgical resection is performed only when endoscopically resected specimens show high-risk features (deep submucosal (SM) invasion, lymphovascular invasion (LVI), tumor budding, or poorly differentiated histology) related to LNM.[7, 11-15] How-ever, even though additional surgery is recommend based on the current guidelines, LNM occurs in 2.1%–25.0% cases treated endoscopically and then surgically. In other words, 75%–98% of additional surgeries are unnecessary. The main challenge is to figure out LNM before underwent surgery. Several studies have attempted to identify a method to predict LNM in patients with T1 CRC (tumor invaded submucosa according to Japanese Society for Cancer of the Colon and Rectum (JSCCR) and American join Com-mittee on Cancer) to reduce the number of unnecessary surgeries and minimize the risk of LNM. However, since low inter-observer agreement and limited indications of current guidelines, it is nearly impossible to predict LNM through pathologic examination based on Hematoxylin and eosin (H&E)-stained endoscopically resected specimen.

Recent studies have attempted to solve these problems using artificial intelligence (AI). The two strategical approaches for AI assisted assessment of the risk of LNM in T1 CRC were pathologist dependent and independent. Pathologist-dependent strategy used text-based data, which included the histologic features obtained by pathologist, such as depth of SM invasion, tumor differentiation, and LVI. These test data- based AI model proved sufficient evidence with large cohorts and ex-ternal validation, outperforming the current guidelines. Nevertheless, there were still limitations, such as varying pathologic criteria and standards among different guidelines and diagnostic disagreement among pathologists. To address these issues, pathologist-independent AI model utilizing WSIs had been reported. WSI-based AI models with hematoxylin and eosin staining alone, including our previous study were simplified and less disruptive than current clinical best practice. This strategy ap-pears to be more ideal to overcome inter-observer discrepancy, but a relatively low area under the curve (AUC) compared to the pathologist-dependent method and external vali-dation remain a challenge.

Even though our previous study demonstrated the potential of using an AI with H&E-stained WSIs from endoscopically resected specimens without handcrafted features to predict LNM in patients with T1 CRC, our model had certain limitations. WSIs from endoscopic resected specimens had high risk histological features of LNM, because it be-longed to patients who underwent additional surgery after endoscopic treatment. So, previous model was unsuitable for predicting LNM in low-risk patients with T1 CRC. Also, study population was small (n=400) and AUC was relatively low. To increase the number of patients and WSIs from patients with low risk of LNM, we conducted a study with AI training and testing by expanding the scope to include previously endoscopically resected specimens from patients who underwent additional surgery due to the high risk of LNM, as well as surgical specimens from patients who underwent surgery for T1 CRC. Since previous model was lack of external validation with an independent cohort, we wanted to perform extensive external validation with WSIs from multi-centers and apply to WSIs from T1 CRC patients who underwent endoscopic treatment. However, it takes a lot of time to prepare WSI from multicenter and obtain 5-year overall survival rate in patients who underwent only endoscopic treatment for T1 CRC. So, we conducted study with alternative method instead of external validation. To validation with independent cohort in single center, we aimed to develop an AI model with four versions comprising various combinations of training and test sets using H&E-stained WSIs from surgical and endoscopically resected specimens. Because, endoscopic resected specimens contained only part of the SM layer while, surgical specimen contained the entire layer of intestine, so two cohorts were independent each other. Additionally, we aimed to apply our AI program to predict LNM in T1 CRC samples.

  1. The conclusion is weak. More detailed informative and guiding information should have been given in this section.

à Author Response: Thank you for your thoughtful comment. Following your advice, we added more detailed information and reinforced the conclusion.

Conclusion section (16 page, line 468-477)

In conclusion, our AI model with H&E-stained WSIs and without pathologists showed higher performance power (AUC, 0.782-0.824) with validation of an independent cohort in a single center than previous studies. Since WSIs from 1281 patients with low to high risk of LNM were used to develop present AI model, it was suitable for predicting LNM even in low-risk patients with T1 CRC. Moreover, this model reduced 14.2% of unnecessary additional surgeries than prediction using the current JSCCR guidelines while allowing 7% of missed LNM. This revealed the feasibility of using an AI model with only H&E-stained WSIs to predict LNM in T1 CRC. However, to apply our model to real-world clinical practice, extensive external validation with WSI from multiple centers and patients who undergo only endoscopic treatment is warranted.

Reviewer 2 Report

Comments and Suggestions for Authors

Review : Prediction of lymph node metastasis ….

General remark :

There is no doubt,that machine learning and AI will become an important role / tool for medical diagnosis.

In this context the paper under review represents a very detailed and comprehensive contributrion to this topic.

Remarks :

1. Quality control of the pathohistologic specimens regarding comparability in respect to thickness of microtomic slices, staining parameters , artifacts etc.) should be discussed.

2. The meaning of submucosal invasion depth is still under controversial discussion both by measurement ( micrometers ) as well as by levels (sm1 –sm3 (4) ). The same holds true for budding.

3. Even the definition of T1-Tumors is in part different in western countries compared to asia and USA.

4. Due to the fact, that the submucosa in endoscopic resected specimen in most cases is only partly represented ( even in ESD ) the comparison with surgical specimen ( with completely preserved  submucosa) may be critical.

4. A decisive factor, which should be mentioned,  is the quality of LN- assessment : Detection and numbers of LN strictly depend on surgical techniques and scrupulousness in gathering and working up. What about comparability esp. over the years ?

5. Some sentences  lack clearness ( e.g. 293-297, 317-319, 321-323, 323-324 etc. ) and are hard to understand.

6. The decision to choose version 4  ( 305-328 ) should be declared more clearly

7. The 0s in table 4 should be explained

 Concerning  statistics, ,DL and AI .

A. Exact meaning and consequences  of several parameters for the final conclusions should be better declared ( e.g. AUC, AS, RF etc. )

B. Tab 1 is in part confusing, as some results are not exspected ( e.g. in  Differentiation, Budding, Pos.resection margin etc.) and not discussed

Formal aspects :

 I.  For the clinical reader, not daily dealing with statistics and AI parameters, some short explanations should be given ( e.g. Youden index, McNemar´s test ) and a compact list of the abundant abbreviations  seems urgently needed.

II. Tab.1 should be printed in a more clearly arranged form to avoid confusion with the lines.

III.  Fig.3 : Nobody can read these tiny typefaces without a magnifier !!

IV. Fig.4 : the same as Fig.3

Author Response

Point-by-point response

We sincerely thank you for the opportunity to revise our manuscript entitled “Prediction of lymph node metastasis in T1 colorectal cancer using artificial intelligence with hematoxylin and eosin-stained whole slide images of endoscopic and surgical resection specimens" (cancers-2980766) to allow it to be reconsidered for publication in Cancers. We have revised the manuscript based on the reviewers’ helpful comments and recommendations. We have done our best to address to these comments thoroughly, and our responses are as follows:

Answers to comments from the Editorial team:

Reviewer #2

Remarks:

  1. Quality control of the pathohistological specimens regarding comparability in respect to thickness of microtomic slices, staining parameters, artifacts etc.) should be discussed.

à Author Response: Thank you for your thoughtful comment. Following your advice, we added following test in methods section.

Methods section (4 page, lines 151-159)

Assessment of histological differentiation was based on the least high-grade pattern of the carcinoma which often co-exist with dominant elements of low-grade patterns. Immunostaining with D2–40 was occasionally performed for lymphatic vessel to determine whether it was a true lymphatic or an iatrogenic empty space caused by tissue being pushed in the process of specimen fixing in formaldehyde and making into slide.

Surgical and endoscopically resected specimens were fixed in formaldehyde and embedded in paraffin. Tissue specimens were cut into sections with 3μm that were placed on the slides. During preparation, the artifact was removed from ethanol and 50⁰C floating hot water tank.

  1. The meaning of submucosal invasion depth is still under controversial discussion both by measurement (micrometers) as well as by levels (sm1 –sm3 (4)). The same holds true for budding.

à Author Response: Thank you for your considerable comment.

In T1 CRC, it was often difficult to measure depth of SM invasion. Also, there was inter-observer variation for measurement. As you mentioned, the meaning of depth of invasion on LNM is controversy. According to JSCCR guidelines, SM depth >1000μm was one of the risk factors for LNM, but several studies showed difference in SM invasion depth related to LNM. Nakadoi et al. (J Gastroenterol Hepatol. 2012;27(6)1057-62) demonstrated that even when only three conditions were considered regardless of the depth of submucosal invasion, i.e. absence of vascular invasion, grade 1 tumor budding, and well-/moderately-differentiated/papillary histology, the incidence of LN metastasis was reduced to 1.2%. Tanaka et al. (Oncol Rep. 2000;7(4):783-791) indicated that CRC with SM invasion<1500μm could be cured by complete EMR on conditions that histologic grade at the deepest invasive portion is well or moderate-to well, if there are no vessel involvement. Egashira et al. (Modern Pathology. 2004;17(5):503-511) noted that by scoring the risk factors for LN metastasis, i.e. venous invasion, lymphocytic infiltration, submucosal invasion (≥2000µm), cribriform structure, high-grade tumor budding, and mucinous histologic differentiation, it becomes possible to reduce the number of unnecessary additional surgeries.

As you mentioned, the meaning of tumor budding for LNM in CRC is also controversy. The International Tumor Budding Consensus Conference (ITBCC) guidelines recommended the use of a three-tier system for risk stratification: Bd 1, low budding (0-4 buds); Bd 2, intermediate budding (5-9 buds); Bd 3, high budding (10 or more buds). However, there was inconsistency in tumor bud counts among pathologists. In p1 CRC, Bd 2 and Bd 3 were associated with an increased risk of LNM, whereas in stage II CRC, Bd 3 is associated with an increased risk of recurrence and mortality. Moreover, the current tumor bud assessment system focused only on the tumor bud count and did not account for other features (structure, location, cell atypia, stroma type and tumor bud cell mitosis etc.). Ozeki et al. study analyzing LNM for only high risk pT1 CRC, revealed that no LNMs were observed in pT1 CRCs with a negative vertical margin and SM invasion ≤2000µm that had no other risk factors except for budding. (Cancers (Basel). 2022;14(3)) Budding did not add the risk of LNM (4.2%) in pT1 CRC with invasion depth ≥1000 µm but raised the risk of LNM (21.4%) in pT1 CRC with both risk factors of SM invasion depth and lymphovascular invasion. They suggested new criteria to omit additional surgery in patients with high risk pT1 CRC who met all five criteria: 1) differentiated; 2) no LVI; 3) colon cancer; 4) SM invasion depth ≤2000µm; and (5) negative vertical margin.

In pathologist-dependent AI model, the depth of SM invasion and tumor budding were one of text-based data, obtained by pathologist. So, we wanted to figure out the meaning of depth of invasion for LNM and tumor budding without pathologist annotation, as pathologist-independent strategy. We developed an attention-based WSI-level classification deep learning model to predict whether a WSI is LNM positive or negative. As a results, in our study, the prominent features of the patch images with high ASs were tumor budding and micropapillary patterns.

Following your advice, we added following text in discussion section.

Discussion section (15 page, line 432-444)

Even though, depth of SM invasion and tumor budding were risk factors of LNM, the meaning of these is controversy and inter-observer variation for measurement was existed. According to JSCCR guidelines, SM depth >1000μm was one of the risk factors for LNM, but several studies showed difference in SM invasion depth related to LNM.[43-45] The International Tumor Budding Consensus Conference guidelines recommended the use of a three-tier system for risk stratification: Bd 1, low budding (0-4 buds); Bd 2, intermediate budding (5-9 buds); Bd 3, high budding (10 or more buds).[46] In p1 CRC, Bd 2 and Bd 3 were associated with an increased risk of LNM, whereas in stage II CRC, Bd 3 is associat-ed with an increased risk of recurrence and mortality. Moreover, the current tumor bud assessment system focused only on the tumor bud count and did not account for other features.[47] . So, we wanted to figure out the meaning of depth of invasion for LNM and tumor budding using attention-based WSI-level classification deep learning model.

  1. Even the definition of T1-Tumors is in part different in western countries compared to Asia and USA.

à Author Response: Thank you for your kind comment. In our study, T1 tumor was defined that tumor invaded submucosa according to JSCCR and AJCC. Following your advice, we added following text to the introduction section.

Introduction section (3page, line 83-86)

Several studies have attempted to identify a method to predict LNM in patients with T1 CRC (tumor invaded submucosa according to Japanese Society for Cancer of the Colon and Rectum (JSCCR) and American join Committee on Cancer) to reduce the number of unnecessary surgeries and minimize the risk of LNM.

  1. Due to the fact, that the submucosa in endoscopic resected specimen in most cases is only partly represented (even in ESD) the comparison with surgical specimen (with completely preserved submucosa) may be critical.

à Author Response: Thank you for your considerable comment. As you mentioned, endoscopic resected specimens contained only part of the SM layer while, surgical specimen contained the entire layer of intestine. So endoscopic resected specimen and surgical specimen were different and two cohort were independent each other.  It is the reason that we developed a model with four versions comprising various combinations of training and test sets using H&E-stained WSIs from endoscopically (400 patients) and surgically resected specimens (881 patients). Since, our final goal is to apply AI model to only endoscopic treated T1 CRC patients who did not need additional surgery, we actually did not perform direct comparison WSIs between endoscopic resected specimen and surgical specimen. Instead, we modified the Table 1 to compare clinical and pathological features between two specimens.

Table 1 Table 1. Baseline characteristics of study population.

Endoscopic resection followed by additional surgery (n=400)

Surgical resection

(n=881)

Total

Negative LNM

(n=329)

Positive LNM

(N=71)

Negative LNM

(n=780)

Positive LNM

(N=101)

p

value*

Clinical features

Age at diagnosis

Year (IQR)

60.0 (52.0–68.0)

59.0 (52.0–65.0)

60.0 (52.0–68.0)

60.0 (52.0–69.0)

59.0 (52.0–67.0)

0.041

Sex

Male

Female

764 (59.6) 517 (40.4)

193 (58.7)

136 (41.3)

46 (64.8)

25 (35.2)

467 (59.9)

313 (40.1)

58 (57.4)

43 (42.6)

0.504

Body mass index

kg/m2 (IQR)

24.1 (22.2-26.1)

23.9 (22.0–26.1)

24.8 (23.3–27.4)

24.1 (22.2–26.0)

24.8 (23.0–26.0)

0.704

Presence of

comorbidity

No

Yes

795 (62.1)

486 (37.9)

217 (66.0)

112 (34.0)

36 (50.7)

35 (49.3)

477 (61.2)

302 (38.7)

65 (64.4)

36 (35.6)

0.299

Family history of CRC

No

yes

1143 (89.2)

138 (10.8)

294 (86.3)

45 (13.7)

61 (85.9)

10 (14.1)

708 (90.8)

72 (9.2)

90 (89.1)

11 (10.9)

0.014

Smoking status

No

Ex-smoker

Yes

912 (71.2)

201 (15.7)

168 (13.1)

214 (65.0)

59 (17.9)

56 (17.0)

43 (60.6)

10 (14.1)

18 (25.4)

588 (75.4)

104 (13.3)

88 (11.3)

67 (66.3)

23 (22.8)

11 (10.9)

<0.001

Alcohol consumption

No

Ex-drinker

Yes

809 (63.2)

71 (5.5)

401 (31.3)

192 (58.4)

27 (8.2)

110 (33.4)

31 (43.7)

9 (12.7)

31 (43.7)

523 (67.1)

29 (3.7)

228 (29.2)

63 (62.4)

6 (5.9)

32 (21.7)

<0.001

Tumor location

Left side

Right side

913 (71.3)

368 (28.7)

241 (73.3)

88 (26.7)

50 (70.4)

21 (29.6)

542 (69.5)

238 (30.5)

80 (79.2)

21 (20.8)

0.236

Pathologic features

Size of cancer

mm (IQR)

15.0 (10.0-22.0)

10.0 (7.3–14.0)

8.0 (7.0–12.0)

20.0 (15.0–25.0)

16.5 (14.3–25.0)

<0.001

Depth of

SM invasion

μm (IQR)

1775.0 (1000.0-2200.0)

1800.0 (1075.0–2300.0)

1500.0

(1000.0–2000.0)

N/A

SM1

340 (38.6)

318 (40.8)

22 (21.8)

N/A

SM2

218 (24.7)

185 (23.7)

33 (32.7)

SM3

323 (36.7)

277 (35.5)

46 (45.5)

Differentiation

Well

760 (59.3)

182 (55.3)

55 (77.5)

480 (61.5)

43 (42.6)

0.210

Moderate

485 (37.9)

133 (40.4)

14 (19.7)

286 (36.7)

52 (51.5)

Poorly

36 (2.8)

14 (4.3)

2 (2.8)

14 (1.8)

6 (5.9)

Lympho-vascular invasion

No

Yes

1030 (80.4)   251 (19.6)

243 (73.9)

86 (26.1)

49 (69.0)

22 (31.0)

691 (89.6)

89 (11.4)

47 (46.5)

54 (53.5)

<0.001

Tumor budding

No

Yes

1084 (84.6)

197 (15.4)

289 (87.8)

40 (12.2)

62 (87.3)

9 (12.7)

667 (85.5)

113 (14.5)

66 (65.3)

35 (34.7)

0.021

Positive resection margin

No

Yes

1167 (91.1)

114 (8.9)

235 (71.4)

94 (28.6)

51 (71.8)

20 (28.2)

780 (100)

0

101 (100)

0

<0.001

Microsatellite stability

Stable

Unstable

Unknown

915 (71.5)

86 (6.7)

279 (21.8)

82 (25.0)

8 (2.4)

238 (72.6)

28 (39.4)

2 (2.8)

41 (57.7)

667 (85.5)

69 (8.8)

44 (5.6)

89 (88.1)

7 (6.9)

5 (5.0)

<0.001

*p value : differecne bewteen endoscopic resection followed by additional surgery (n=400) and surgical resection (n=881)

LN, lymph node; IQR, interquartile range; CRC, colorectal cancer; SM, submucosal; N/A; not applicable; SM1, upper third, if depth of submucosal invasion was pragmatically divided in equal thirds accoirding to Kudo classification; SM2, middle third; SM3, lower third

  1. A decisive factor, which should be mentioned, is the quality of LN- assessment: Detection and numbers of LN strictly depend on surgical techniques and scrupulousness in gathering and working up. What about comparability esp. over the years?

à Author Response: Thank you for your thoughtful comment. We only mentioned LN yield and LN ratio in cases of surgical specimens, so we added additional data about in cases of endoscopic resected specimens followed by surgery. LN yield, total number of LNs retrieved after surgery was 22022. And LN ratio, the ratio of positive LNs out of the total removed was 1.24% (273/22022). The AJCC/UICC recommend a minimum of 12 LNs should be identified in colorectal cancer specimens. In our study, average of 17 LNs were retrieved in each surgery. When we compared past group (patient who underwent surgery in 2003-2010) and recent group (2011-2020), average of 16 LNs were retrieved per surgery in past group, and 18 LNs were retrieved per surgery in recent group. In summary, our study sufficiently met the minimal LN assessment regardless of the year of surgery, and recently more LNs were retrieved in each surgery. Following your advice, we have added following text to the Results section.

Results section (6 page, line 232-237)

LN yield, total number of LNs retrieved after surgery was 22022. And LN ratio, the ratio of positive LNs out of the total removed was 1.24 % (273/22022). In our study, average of 17 LNs were retrieved in each surgery. When we compared past group (patient who under-went surgery in 2023-2010) and recent group (2011-2020), average of 16 LNs were retrieved per surgery in past group, and 18 LNs were retrieved per surgery in recent group.

  1. Some sentences lack clearness (e.g. 293-297, 317-319, 321-323, 323-324 etc.) and are hard to understand.

à Author Response: Thank you for your kind comment. Following your advice, we have modified these sentences.

Line 364-370

To increase the number of patients and WSIs from patients with low risk of LNM, we included patients who underwent surgery between 2003–2020, a longer study period than those who underwent endoscopic resection followed by surgery (2010–2018). Massive surgical specimens that performed when surgery was the only treatment option for T1 CRC, showed lower risk of LNM, compared to endoscopic resected specimens.

Line 396-399

In previous study, when we compared AI model with JSCCR guidelines, we ideally set the cutoff threshold of the AI model at 100% sensitivity not to allow missed LNM, like JSCCR guidelines. As a results, previous model reduced unnecessary additional surgeries by 15.1% than the current JSCCR guidelines.

Line 401-406

Present model avoided 14.2-45.8% of unnecessary additional surgeries than prediction using the current JSCCR guidelines, while allowing missed LNM, ranged to 7.1-28.6%.

Line 403-404

The AI model that reduced more unnecessary additional surgery allowed more missed LNM.

  1. The decision to choose version 4 ( 305-328 ) should be declared more clearly

à Author Response: Thank you for your kind comment. Version 4 (train: endoscopic and surgical specimens, test: endoscopic specimens) was the closest to the actual prediction target (patient who only treated with endoscopic treatment for T1 CRC), and showed the greatest improvement of AUC (0.265), compared to RF with clinicopathologic features, among four versions. Moreover, it allowed acceptable the lowest rate of missed LNM to reduced unnecessary additional surgery. Considering these points, we though that AI model with Version 4 was compatible for clinical practice to predict LNM in T1 CRC.

Following your advice, we have modified these sentences.

Discussion section (14-15 page, line 373-408)

We developed an AI model with a DL program to predict LNM in T1 CRC using surgical and endoscopic specimens. The AUCs of our DL model were 0.758–0.830 for the training set and 0.781–0.824 for the test set; which were improved compared with the previous model (AUC: 0.747 in training and 0.767 in testing sets). This could be explained by the fact that surgical specimens contained much more tissue and information than endoscopically resected specimens, and over 1000 WSIs were used for analysis compared with previous studies. Among four versions, Version 3 (train: endoscopic and surgical specimens, test: surgical specimens) which contained surgical specimens in both training and test sets had highest performance power. However, all versions of the AI model showed acceptable AUC ranges for predicting LNM in patients with T1 CRC. On the other hand, RF with clinicopathological features showed a lower AUC (0.516–0.701) than the AI mod-el. Only Version 1 (training and test: surgical specimen) had an AUC barely greater than 0.7.

Among four versions, Version 4 (train: endoscopic and surgical specimens, test: endoscopic specimens) was the closest to the actual prediction target. Because the ultimate goal of the AI model was to predict LNM in patient who only treated with endoscopic treatment for T1 CRC. However, considering the study results, it could be assumed that test with endoscopic resected specimen was difficult to predict LNM. The AUC of RF with clinicopathologic features was the lowest (0.516) in Version 4. And the AUC of Version 4 was the lowest among the four versions, even though the differenced was not significant. However, it was remarkable that AI model with Version 4 had the greatest improvement of AUC (0.265), compared to RF with clinicopathologic features, among four versions. Thus, our newly developed model showed the possibility of application in clinical practice for LNM prediction.

In previous study, when we compared AI model with JSCCR guidelines, we ideally set the cutoff threshold of the AI model at 100% sensitivity not to allow missed LNM, like JSCCR guidelines. As a results, previous model reduced unnecessary additional surgeries by 15.1% than the current JSCCR guidelines. However, setting the cutoff threshold at 100% sensitivity might not reflect reality, we used Youden index for setting the cutoff in our pre-sent study. Present model avoided 14.2-45.8% of unnecessary additional surgeries than prediction using the current JSCCR guidelines, while allowing missed LNM, ranged to 7.1-28.6%. The AI model that reduced more unnecessary additional surgery allowed more missed LNM. So careful interpretation of results was needed. Considering, acceptable the lowest rate of missed LNM, ultimate target population, and improvement of AUC using AI model, AI model with Version 4 was compatible for clinical practice to predict LNM in T1 CRC. It reduced 14.2% of unnecessary additional surgeries than prediction using the current JSCCR guidelines while allowing 7% of missed LNM.

  1. The 0s in table 4 should be explained

à Author Response: Thank you for your comment. JSCCR guidelines recommend that additional colorectal surgery was performed, when endoscopically resected specimens showed at least one of high-risk features of LNM (positive resection margin, deep SM invasion, presence of LVI, poorly differentiated histology, or tumor budding. JSCCR guideline did not allow any LNM of T1 CRC, while, resulted in unnecessary additional surgery. It meant that this strategy showed 100% sensitivity and 0% specificity. Previous study had attempted this kind of comparison. See below (table in Ichimasa et al. study, Endoscopy.2018;50(3):230-240)

Following your advice, we have added explanation in results section.  

Results section (11 page, line 308-313)

We compared the performance of our model (four versions) with that of JSCCR guidelines using the test set (Table 4). JSCCR guidelines recommend that additional colorectal surgery was performed, when endoscopically resected specimens showed at least one of high-risk features of LNM. JSCCR guideline did not allow any LNM of T1 CRC, while, resulted in unnecessary additional surgery. It meant that this strategy showed 100% sensitivity and 0% specificity.

 Concerning statistics, DL and AI.

  1. Exact meaning and consequences of several parameters for the final conclusions should be better declared ( e.g. AUC, AS, RF etc. )

à Author Response: Thank you for your kind comment. Following your advice, we added more detailed meaning and consequence of several parameters in manuscript.

Methods section (5 page, lines 192-195)

The AI performance was evaluated using the AUC receiver operating characteristic curve (ROC). ROC is a probability curve and AUC represents the degree or measure of separability. It showed how much the model was capable of distinguishing between classes.

Methods section (5 page, line 176-180)

The AM computes an attention score (AS), between 0–1, for each patch image in a WSI; the sum of these scores is equal to 1. An attention mechanism was used to visualize the spatial distribution of ASs of the WSIs. A higher AS indicates that the patch image is relatively more informative and has a greater influence on the final classification decision.

Methods section (6 page, line 205-208)

By comparing our AI model with a model using clinicopathological features, we trained a random forest (RF) classifier with 500 trees to predict LNM.[35] RF is versatile and widely used machine learning algorithm that constructed multiple decision trees and combined their outputs for robust and accurate predictions.

  1. Tab 1 is in part confusing, as some results are not expected ( e.g. in  Differentiation, Budding, Pos.resection margin etc.) and not discussed

à Author Response: Thank you for your thoughtful comment. We additionally wanted to analyze pathologic features related to high risk of LNM. Following your advice, we modified Table 1, and discussed some variables with statistically significant results.

Results section (6 page, line 221-229)

The median age at CRC diagnosis was much younger in patients with endoscopic resection followed by additional surgery (59.0; IQR, 52.0–65.0), than in patients with surgical resection (60.0; IQR, 52.0–69.0). Men accounted for 59.6% of the total population. The per-centage of patient with a family history of CRC, ex-/current smoker, or alcohol ex-/current drinker was higher in endoscopic resection followed by additional surgery. Patients without a family history of CRC accounted for 89.2% of the patients. High risk pathologic features related to LNM, including LVI, tumor budding, positive resection margin, and microsatellite instability was more in patients with endoscopic resection followed by additional surgery, than surgical resection.

Discussion section (14 page, line 367-370)

Massive surgical specimens that performed when surgery was the only treatment option for T1 CRC, showed lower risk of LNM, compared to endoscopic resected specimens. In-deed, pathologic characteristics of patients with surgical resection were lower risk of LNM than in patients with endoscopic resection, followed by surgery.

Formal aspects:

  1. For the clinical reader, not daily dealing with statistics and AI parameters, some short explanations should be given ( e.g. Youden index, McNemar´s test) and a compact list of the abundant abbreviations  seems urgently needed.

à Author Response: Thank you for your kind comment. Following your advice, we added explanations for statistical methods and provided a list of the abbreviations.

Methods section (6 page, line 209-214)

The optimal cut-off sensitivity and specificity of each model were evaluated using the Youden index, the maximum potential effectiveness of a diagnostic biomarker, and a common summary measure of the ROC curve.[36] And we used McNemar’s tests, non-parametric test used to analyze paired nominal data, to compare predictive performances between our model and JSCCR guidelines, the most widely used guidelines in Asia.

List of abbreviations (2 page line 48-69)

CRC, colorectal cancer

LNM, lymph node metastasis

SM, submucosal

LVI, lymphovascular invasion

JSCCR, Japanese Society for Cancer of the Colon and Rectum

H&E, Hematoxylin and eosin

AI, artificial intelligence

WSI, whole slide images

AUC, area under the curve

EMR, endoscopic mucosal resection

ESD, endoscopic submucosal dissection

DCNN, a deep convolutional neural network

AM, attention module

CM, classification module

AS, attention score

FV, feature vector

IQR, interquartile ranges

ROC, receiver operating characteristic curve

CV, cross-validation

RF, random forest

ROI, regions of interest

  1. Tab.1 should be printed in a more clearly arranged form to avoid confusion with the lines.

à Author Response: Thank you for your comment. Following your advice, we modified Table 1.

Table 1. Baseline characteristics of study population.

Endoscopic resection followed by additional surgery (n=400)

Surgical resection

(n=881)

Total

Negative LNM

(n=329)

Positive LNM

(N=71)

Negative LNM

(n=780)

Positive LNM

(N=101)

p

value*

Clinical features

Age at diagnosis

Year (IQR)

60.0 (52.0–68.0)

59.0 (52.0–65.0)

60.0 (52.0–68.0)

60.0 (52.0–69.0)

59.0 (52.0–67.0)

0.041

Sex

Male

Female

764 (59.6) 517 (40.4)

193 (58.7)

136 (41.3)

46 (64.8)

25 (35.2)

467 (59.9)

313 (40.1)

58 (57.4)

43 (42.6)

0.504

Body mass index

kg/m2 (IQR)

24.1 (22.2-26.1)

23.9 (22.0–26.1)

24.8 (23.3–27.4)

24.1 (22.2–26.0)

24.8 (23.0–26.0)

0.704

Presence of

comorbidity

No

Yes

795 (62.1)

486 (37.9)

217 (66.0)

112 (34.0)

36 (50.7)

35 (49.3)

477 (61.2)

302 (38.7)

65 (64.4)

36 (35.6)

0.299

Family history of CRC

No

yes

1143 (89.2)

138 (10.8)

294 (86.3)

45 (13.7)

61 (85.9)

10 (14.1)

708 (90.8)

72 (9.2)

90 (89.1)

11 (10.9)

0.014

Smoking status

No

Ex-smoker

Yes

912 (71.2)

201 (15.7)

168 (13.1)

214 (65.0)

59 (17.9)

56 (17.0)

43 (60.6)

10 (14.1)

18 (25.4)

588 (75.4)

104 (13.3)

88 (11.3)

67 (66.3)

23 (22.8)

11 (10.9)

<0.001

Alcohol consumption

No

Ex-drinker

Yes

809 (63.2)

71 (5.5)

401 (31.3)

192 (58.4)

27 (8.2)

110 (33.4)

31 (43.7)

9 (12.7)

31 (43.7)

523 (67.1)

29 (3.7)

228 (29.2)

63 (62.4)

6 (5.9)

32 (21.7)

<0.001

Tumor location

Left side

Right side

913 (71.3)

368 (28.7)

241 (73.3)

88 (26.7)

50 (70.4)

21 (29.6)

542 (69.5)

238 (30.5)

80 (79.2)

21 (20.8)

0.236

Pathologic features

Size of cancer

mm (IQR)

15.0 (10.0-22.0)

10.0 (7.3–14.0)

8.0 (7.0–12.0)

20.0 (15.0–25.0)

16.5 (14.3–25.0)

<0.001

Depth of

SM invasion

μm (IQR)

1775.0 (1000.0-2200.0)

1800.0 (1075.0–2300.0)

1500.0

(1000.0–2000.0)

N/A

SM1

340 (38.6)

318 (40.8)

22 (21.8)

N/A

SM2

218 (24.7)

185 (23.7)

33 (32.7)

SM3

323 (36.7)

277 (35.5)

46 (45.5)

Differentiation

Well

760 (59.3)

182 (55.3)

55 (77.5)

480 (61.5)

43 (42.6)

0.210

Moderate

485 (37.9)

133 (40.4)

14 (19.7)

286 (36.7)

52 (51.5)

Poorly

36 (2.8)

14 (4.3)

2 (2.8)

14 (1.8)

6 (5.9)

Lympho-vascular invasion

No

Yes

1030 (80.4)   251 (19.6)

243 (73.9)

86 (26.1)

49 (69.0)

22 (31.0)

691 (89.6)

89 (11.4)

47 (46.5)

54 (53.5)

<0.001

Tumor budding

No

Yes

1084 (84.6)

197 (15.4)

289 (87.8)

40 (12.2)

62 (87.3)

9 (12.7)

667 (85.5)

113 (14.5)

66 (65.3)

35 (34.7)

0.021

Positive resection margin

No

Yes

1167 (91.1)

114 (8.9)

235 (71.4)

94 (28.6)

51 (71.8)

20 (28.2)

780 (100)

0

101 (100)

0

<0.001

Microsatellite stability

Stable

Unstable

Unknown

915 (71.5)

86 (6.7)

279 (21.8)

82 (25.0)

8 (2.4)

238 (72.6)

28 (39.4)

2 (2.8)

41 (57.7)

667 (85.5)

69 (8.8)

44 (5.6)

89 (88.1)

7 (6.9)

5 (5.0)

<0.001

*p value : differecne bewteen endoscopic resection followed by additional surgery (n=400) and surgical resection (n=881)

LN, lymph node; IQR, interquartile range; CRC, colorectal cancer; SM, submucosal; N/A; not applicable; SM1, upper third, if depth of submucosal invasion was pragmatically divided in equal thirds according to Kudo classification; SM2, midel third; SM3, lower third

III.  Fig.3 : Nobody can read these tiny typefaces without a magnifier !!

à Author Response: Thank you for your comment. Following your advice, we modified Figure 3.

  1. Fig.4 : the same as Fig.3

à Author Response: Thank you for your comment. Following your advice, we modified Figure 4.

Reviewer 3 Report

Comments and Suggestions for Authors

The authors proposed an AI-based model that allows one to predict the presence of metastasis in the lymph nodes and thereby reduce the number of unnecessary surgical interventions. The idea of the manuscript seems quite interesting to me; the results have the potential for implementation in clinical practice. I think the contribution of this manuscript to the field of study is quite high.
I didn’t like the style of presentation; it’s difficult to understand what the authors are writing about, what principle is used to form the variants 1-4, etc. This was one of the main questions regarding the methodology - the rationale for separating subgroups into test and training (variants 1-4). I indicated in the comments earlier. The conclusion is justified in principle and is consistent with the results obtained. The authors have a fairly high number of matches for anti-plagiarism, but this may be due to the fact that even within this manuscript they repeat the same phrases many times. However, work still needs to be done to reduce repetition.

1. Delete lines 159-161.

2. Explain in Table 1 SM1, SM2, SM3.

3. In general, Table 1 is not good. Are p-values only for comparing negative and positive LNM? no comparison between endoscopic and surgical resection? What do the identified differences mean? Why do the authors provide this table if this is not discussed in any way in the text?

4. There needs to be some justification for why the samples are distributed this way in versions 1-4. It is completely unclear why this is so.

5. From the data in Table 4, it is not entirely clear where 14.2% of unnecessary additional surgeries came from. Clarify please.

Author Response

Point-by-point response

We sincerely thank you for the opportunity to revise our manuscript entitled “Prediction of lymph node metastasis in T1 colorectal cancer using artificial intelligence with hematoxylin and eosin-stained whole slide images of endoscopic and surgical resection specimens" (cancers-2980766) to allow it to be reconsidered for publication in Cancers. We have revised the manuscript based on the reviewers’ helpful comments and recommendations. We have done our best to address to these comments thoroughly, and our responses are as follows:

Answers to comments from the Editorial team:

Reviewer #3:

  1. Delete lines 159-161.

à Author Response: Thank you for your considerable comment. Follow your advice, we deleted lines 159-161.

  1. Explain in Table 1 SM1, SM2, SM3.

à Author Response: Thank you for your kind comment. Depth of submucosal invasion, a risk factor for lymph node metastasis was evaluated according to Kudo classification (SM1, SM2, or SM3).(Kudo, Tamegai et al. 2001) If the invasion depth into the submucosa is pragmatically divided in equal thirds, upper third is SM1, middle third is SM2, and lower third is SM3. Deep submucosal invasion was in case of depth of invasion ≥ 1000μm or SM2-3.

Following your advice, we mentioned the meanings of SM1, SM2, and SM3 in Table 1.

  1. In general, Table 1 is not good. Are p-values only for comparing negative and positive LNM? no comparison between endoscopic and surgical resection? What do the identified differences mean? Why do the authors provide this table if this is not discussed in any way in the text?

à Author Response: Thank you for your thoughtful comment. Following your advice, we performed comparison analysis between endoscopic resection followed by additional surgery (n=400) and surgical resection (n=881), and modified table 1. And we added following text in results section.

Table 1. Baseline characteristics of study population.

Endoscopic resection followed by additional surgery (n=400)

Surgical resection

(n=881)

Total

Negative LNM

(n=329)

Positive LNM

(N=71)

Negative LNM

(n=780)

Positive LNM

(N=101)

p

value*

Clinical features

Age at diagnosis

Year (IQR)

60.0 (52.0–68.0)

59.0 (52.0–65.0)

60.0 (52.0–68.0)

60.0 (52.0–69.0)

59.0 (52.0–67.0)

0.041

Sex

Male

Female

764 (50.59.6) 517 (40.4)

193 (58.7)

136 (41.3)

46 (64.8)

25 (35.2)

467 (59.9)

313 (40.1)

58 (57.4)

43 (42.6)

0.504

Body mass index

kg/m2 (IQR)

24.1 (22.2-26.1)

23.9 (22.0–26.1)

24.8 (23.3–27.4)

24.1 (22.2–26.0)

24.8 (23.0–26.0)

0.704

Presence of

comorbidity

No

Yes

795 (62.1)

486 (37.9)

217 (66.0)

112 (34.0)

36 (50.7)

35 (49.3)

477 (61.2)

302 (38.7)

65 (64.4)

36 (35.6)

0.299

Family history of CRC

No

yes

1143 (89.2)

138 (10.8)

294 (86.3)

45 (13.7)

61 (85.9)

10 (14.1)

708 (90.8)

72 (9.2)

90 (89.1)

11 (10.9)

0.014

Smoking status

No

Ex-smoker

Yes

912 (71.2)

201 (15.7)

168 (13.1)

214 (65.0)

59 (17.9)

56 (17.0)

43 (60.6)

10 (14.1)

18 (25.4)

588 (75.4)

104 (13.3)

88 (11.3)

67 (66.3)

23 (22.8)

11 (10.9)

<0.001

Alcohol consumption

No

Ex-drinker

Yes

809 (63.2)

71 (5.5)

401 (31.3)

192 (58.4)

27 (8.2)

110 (33.4)

31 (43.7)

9 (12.7)

31 (43.7)

523 (67.1)

29 (3.7)

228 (29.2)

63 (62.4)

6 (5.9)

32 (21.7)

<0.001

Tumor location

Left side

Right side

913 (71.3)

368 (28.7)

241 (73.3)

88 (26.7)

50 (70.4)

21 (29.6)

542 (69.5)

238 (30.5)

80 (79.2)

21 (20.8)

0.236

Pathologic features

Size of cancer

mm (IQR)

15.0 (10.0-22.0)

10.0 (7.3–14.0)

8.0 (7.0–12.0)

20.0 (15.0–25.0)

16.5 (14.3–25.0)

<0.001

Depth of

SM invasion

μm (IQR)

1775.0 (1000.0-2200.0)

1800.0 (1075.0–2300.0)

1500.0

(1000.0–2000.0)

N/A

SM1

340 (38.6)

318 (40.8)

22 (21.8)

N/A

SM2

218 (24.7)

185 (23.7)

33 (32.7)

SM3

323 (36.7)

277 (35.5)

46 (45.5)

Differentiation

Well

760 (59.3)

182 (76.8)

55 (23.2)

480 (61.5)

43 (42.6)

0.210

Moderate

485 (37.9)

133 (90.5)

14 (9.5)

286 (36.7)

52 (51.5)

Poorly

36 (2.8)

14 (87.5)

2 (12.5)

14 (1.8)

6 (5.9)

Lympho-vascular invasion

No

Yes

1030 (80.4)   251 (19.6)

243 (83.2)

86 (79.6)

49 (16.8)

22 (20.4)

691 (89.6)

89 (11.4)

47 (46.5)

54 (53.5)

<0.001

Tumor budding

No

Yes

1084 (84.6)

197 (15.4)

289 (82.3)

40 (81.6)

62 (17.7)

9 (18.4)

667 (85.5)

113 (14.5)

66 (65.3)

35 (34.7)

0.021

Positive resection margin

No

Yes

1167 (91.1)

114 (8.9)

235 (82.2)

94 (82.5)

51 (17.8)

20 (17.5)

780 (100)

0

101 (100)

0

<0.001

Microsatellite stability

Stable

Unstable

Unknown

915 (71.5)

86 (6.7)

279 (21.8)

82 (74.5)

8 (80.0)

238 (85.3)

28 (25.5)

2 (20.0)

41 (14.7)

667 (85.5)

69 (8.8)

44 (5.6)

89 (88.1)

7 (6.9)

5 (5.0)

<0.001

*p value : differecne bewteen endoscopic resection followed by additional surgery (n=400) and surgical resection (n=881)

LN, lymph node; IQR, interquartile range; CRC, colorectal cancer; SM, submucosal; N/A; not applicable; SM1, upper third, if depth of submucosal invasion was pragmatically divided in equal thirds according to Kudo classification; SM2, middle third; SM3, lower third

Results section (6 page, line 220-229)

Their baseline clinicopathological characteristics are presented in Table 1. The median age at CRC diagnosis was much younger in patients with endoscopic resection followed by additional surgery (59.0; IQR, 52.0–65.0), than in patients with surgical resection (60.0; IQR, 52.0–69.0). Men accounted for 59.6% of the total population. The percentage of patient with a family history of CRC, ex-/current smoker, or alcohol ex-/current drinker was high-er in endoscopic resection followed by additional surgery. Patients without a family his-tory of CRC accounted for 89.2% of the patients. High risk pathologic features related to LNM, including LVI, tumor budding, positive resection margin, and microsatellite instability was more in patients with endoscopic resection followed by additional surgery, than surgical resection.

  1. There needs to be some justification for why the samples are distributed this way in versions 1-4. It is completely unclear why this is so.

à Author Response: Thank you for your considerable comment. The final goal of our AI study is to apply the model to real-world clinical practice to predict lymph node metastasis (LNM) by minimizing the risk of LNM. To achieving this goal, extensive external validation with whole slide images (WSIs) from multi-centers is required and AI model should be applied to WSI from T1 CRC patients who undergo only endoscopic treatment. However, considering that it took a lot of time to prepare WSI from multicenter and obtain 5-year overall survival rate in patients who underwent only endoscopic treatment for T1 CRC, we thought that conducting the study with alternative method instead of external validation might be helpful and essential for achieving ultimate goal.

We wanted to refine previous model using different training and test methods and independent cohort. Since, our study aimed to develop pathologist-independent AI model using only WSIs, we though that AI model based on similar approach but trained and tested with different WSI (endoscopically resected specimen vs. surgical specimen), was different from previous model. Because the performance of AI model varied depending on which test sets were used, fitting and validation of AI model were always important for developing AI model. We reinforced our model with using WSIs from over 1000 patients and not only endoscopic resected specimen but also surgical specimens. As mentioned at discussion section, endoscopic resected specimens had high risk histological features of LNM, because it belonged to patients who underwent additional surgery after endoscopic treatment. So, previous model was unsuitable for predicting LNM in low-risk patients with T1 CRC. However, since present study included massive surgical specimens that performed when surgery was the only treatment option for T1 CRC, it could be applied to low to high risk T1 CRC. Indeed, patients of previous study showed a relatively higher rate of LNM (17.8%) than previous Japanese studies with long-term outcomes (10.8–12.4%). On the other hand, patients in present study showed 13.4 % of LNM. To validate with independent cohort in single center, we developed an AI model with four versions comprising various combinations of training and sets using WSIs from surgical and endoscopically resected specimens. As a result, present model showed better performance with AUC 0.824 and reduced at least 14.2% of unnecessary additional surgery in clinical setting, not ideal setting (set the cutoff threshold at 100% sensitivity). Therefore, it revealed that possibility of application in clinical practice for LNM prediction. Of course, we have prepared extensive external validation with collecting WSI from multi-center and following up long-term periods with multicenter cohort those who underwent only endoscopic treatment for T1 CRC. Following your advice, we added more detailed explanation why we developed AI model using four versions. 

Introduction sections (3-4page, line 105-127)

Even though our previous study demonstrated the potential of using an AI with H&E-stained WSIs from endoscopically resected specimens without handcrafted features to predict LNM in patients with T1 CRC, our model had certain limitations. WSIs from endoscopic resected specimens had high risk histological features of LNM, because it be-longed to patients who underwent additional surgery after endoscopic treatment. So, previous model was unsuitable for predicting LNM in low-risk patients with T1 CRC. Also, study population was small (n=400) and AUC was relatively low. To increase the number of patients and WSIs from patients with low risk of LNM, we conducted a study with AI training and testing by expanding the scope to include previously endoscopically resected specimens from patients who underwent additional surgery due to the high risk of LNM, as well as surgical specimens from patients who underwent surgery for T1 CRC. Since previous model was lack of external validation with an independent cohort, we wanted to perform extensive external validation with WSIs from multi-centers and apply to WSIs from T1 CRC patients who underwent endoscopic treatment. However, it takes a lot of time to prepare WSI from multicenter and obtain 5-year overall survival rate in patients who underwent only endoscopic treatment for T1 CRC. So, we conducted study with alternative method instead of external validation. To validation with independent cohort in single center, we aimed to develop an AI model with four versions comprising various combinations of training and test sets using H&E-stained WSIs from surgical and endoscopically resected specimens. Because, endoscopic resected specimens contained only part of the SM layer while, surgical specimen contained the entire layer of intestine, so two cohorts were independent each other. Additionally, we aimed to apply our AI program to predict LNM in T1 CRC samples.

  1. From the data in Table 4, it is not entirely clear where 14.2% of unnecessary additional surgeries came from. Clarify please.

à Author Response: Thank you for your kind comment. As we showed at table 4, AI model with four versions reduced unnecessary additional surgery compared to JSCCR guidelines. For example, using AI with version 4 resulted 68.3% of unnecessary additional surgery, while using the current JSCCR guidelines resulted 82.5% of unnecessary additional surgery. So, it meant that our model with version 4 reduced 14.2% of unnecessary additional surgeries (68.3% vs. 82.5%). Following your advice, we added number of reduced percentages of unnecessary additional surgery compared to JSCCR guidelines.

Table 4. Predictive value of our artificial intelligence model with four versions, and JSCCR guideline for lymph node metastasis in patients with T1 colorectal cancer

Artificial

intelligence

JSCCR

p value

Version 1

Sensitivity (%)

71.4

100

<0.001

Specificity (%)

92.9

0

<0.001

PPV (%)

57.7

11.9

<0.001

Accuracy (%)

90.4

11.9

<0.001

Unnecessary additional

Surgery (%)

42.3

88.1

<0.001

Missed LNM (%)

28.6

0

<0.001

Reduced unnecessary additional surgery(%)*

45.8

Version 2

Sensitivity (%)

71.4

100

<0.001

Specificity (%)

84.2

0

<0.001

PPV (%)

41.7

13.6

<0.001

Accuracy (%)

82.5

13.6

<0.001

Unnecessary additional

Surgery (%)

58.3

86.4

<0.001

Missed LNM (%)

28.6

0

<0.001

Reduced unnecessary additional surgery (%)

28.1

Version 3

Sensitivity (%)

76.2

100

<0.001

Specificity (%)

85.9

0

<0.001

PPV (%)

42.1

11.9

<0.001

Accuracy (%)

84.7

11.9

<0.001

Unnecessary additional

Surgery (%)

57.9

88.1

<0.001

Missed LNM (%)

23.8

0

<0.001

Reduced unnecessary additional surgery (%)

30.2

Version 4

Sensitivity (%)

92.9

100

<0.001

Specificity (%)

57.6

0

<0.001

PPV (%)

31.7

17.5

<0.001

Accuracy (%)

63.8

17.5

<0.001

Unnecessary additional

Surgery (%)

68.3

82.5

<0.001

Missed LNM (%)

7.1

0

<0.001

Reduced unnecessary additional surgery (%)

14.2

*Reduced unnecessary additional surgery when using artificial intelligence model, compared to JSCCR guidelines

JSCCR, Japanese Society for Cancer of the Colon and Rectum; Version 1, train and test: surgical specimen; Version 2, train and test: endoscopic and surgical specimen; Version 3, train: endoscopic and surgical specimens and test: surgical specimen; Version 4, train: endoscopic and surgical specimens and test: endoscopic specimen; PPV, positive predictive value; LNM, lymph node metastasis

Round 2

Reviewer 1 Report

Comments and Suggestions for Authors

I thank the authors for their detailed explanations. The importance of the study and its contribution to the literature is shown. The requested changes have been made and I think the article can be published in its current form.

Reviewer 2 Report

Comments and Suggestions for Authors

Paper is now ready for publication

Reviewer 3 Report

Comments and Suggestions for Authors

I have no further comments on the manuscript. The authors provided detailed responses to the reviewer’s comments and significantly revised the manuscript. I believe that in its present form the article can be recommended for publication.